# Honeys as Possible Sources of Cholinesterase Inhibitors

**DOI:** 10.3390/nu14142969

**Published:** 2022-07-20

**Authors:** Dominik Szwajgier, Ewa Baranowska-Wójcik, Anna Winiarska-Mieczan, Dorota Gajowniczek-Ałasa

**Affiliations:** 1Department of Biotechnology, Microbiology and Human Nutrition, University of Life Sciences in Lublin, 20-704 Lublin, Poland; dorota.gajowniczek@up.lublin.pl; 2Department of Bromatology and Nutrition Physiology, Institute of Animal Nutrition and Bromatology, University of Life Sciences in Lublin, 20-950 Lublin, Poland; anna.mieczan@up.lublin.pl

**Keywords:** acetylcholine, AChE, BChE, honey, Alzheimer’s disease

## Abstract

Alzheimer’s disease (AD) is a progressive neurodegenerative disease characterised by low levels of the neurotransmitter (acetylcholine), oxidative stress, and inflammation of the central nervous system. The only currently available form of treatment entails the administration of AChE/BChE (acetylcholinesterase/butyrylcholinesterase) inhibitors to patients diagnosed with the disease. However, AD prevention is possible by administering the correct inhibitors with food. The aim of this study was to examine 19 types of honey in terms of their contents of cholinesterase inhibitors. The inhibition of AChE and BChE relative to the respective honey samples was evaluated using Ellman’s colorimetric method, including the “false-positive” effect. The highest potential for AChE inhibition was observed in the case of thyme honey (21.17% inhibition), while goldenrod honey showed the highest capacity for BChE inhibition (33.89%). Our study showed that honeys may provide a rich source of cholinesterase inhibitors and, in this way, play a significant role in AD.

## 1. Introduction

Dementia in recent years has been the fastest spreading disease in the world. It is estimated that approximately 36 million people suffer from the condition, particularly AD (Alzheimer’s disease) [1]. As indicated by research, by the year 2050, 100 million people worldwide are expected to struggle with such problems, which constitutes a threefold increase in the prevalence of the disease relative to the presently recorded numbers of [2].

The development of AD is caused by depletion of the acetylcholine neurotransmitter (ACh) due to acetylcholinesterase (AChE) [3,4]. AChE and butylcholinesterase (BChE) hydrolyse ACh within the synaptic cleft [5]; therefore, one of the possible therapeutic strategies entails the use of AChE [6,7], which, according to the cholinergic theory, results in the elevation of the acetylcholine concentration in the brain and, therefore, improvement of the cognitive function in AD patients [8,9]. The currently available drugs (including cholinesterase inhibitors: rivastigmine, galanthamine, and donepezil) (Figure 1), are included at later stages of the disease’s progression [10]. However, the medicines have some limitations: the improvement of cholinergic activity can last for up to 3 years only, and although the substances are efficient in the symptomatic phase of AD, these compounds have a low effect on the course of AD [11].

One of the potential sources of new inhibitors currently under consideration is honey. Polyphenols present in various honeys have been repeatedly pointed out as the main compounds responsible for the antioxidant, anti-inflammatory, antihyperlipidemic, and antimicrobial properties of this natural product. The molecular pathway of specific actions of polyphenols in cardiovascular diseases, diabetes, neurodegenerative, cancer, and other diseases has been comprehensively discussed in some excellent works [13,14,15]. It has also been demonstrated that honey may play a key role in memory enhancement and may also be used as an agent preventing dementia [16].

In the excellent review of Othman et al. [17], works confirming the positive effects of the consumption of various types of honeys by patients (prevention of cognitive decline and dementia) or animals (improvement of short- and long-term spatial memory, reduction in anxiety, improvement of cognition associated with aging, antidepressant effects, and attenuation of ischemia-induced cognitive impairments) were discussed. Taking as an example Tualang honey produced from Koompassia excels trees, Othman et al. [17] reported on the positive role of honey (its high antioxidant activity, amelioration of the brain areas responsible for memory, oxidative stress reduction, increased levels of the brain-derived neurotrophic factor, and improved acetylcholine concentrations, along with the reduction of AChE activity in the brain. The authors attributed the positive effects to the antioxidant properties of Tualang honey, mainly to the presence of a considerable number of flavonoids and phenolic acids, reaching 83.96 ± 4.53 mg gallic acid equivalents (GAE) per 100 g [17]. Indeed, flavonoids and phenolic acids have been repeatedly detected at considerable levels in various honeys [18].

Taking into consideration the above-cited papers, we evaluated the anti-AChE and anti-BChE activity and the total polyphenolic content (TPC) of 19 honeys. Many individual polyphenols detected in honeys were previously pointed out as efficient cholinesterase inhibitors (e.g., flavonoids and phenolic acids by Szwajgier) [19,20,21].

## 2. Materials and Methods

### 2.1. Chemicals

The reagents were purchased from Sigma-Aldrich (St. Louis, MO, USA), as essentially described in our previous publication [22].

### 2.2. Material

The honeys were purchased in 2019 from stores in Lublin, Poland (Table 1). In the case of each sample, the dry weight was determined after drying, as previously reported [23].

### 2.3. Determination of ChE Inhibitory Activity

The inhibition of AChE was tested based on the method of Ellman et al. [24] with some modifications, given in our previous publication [22].

### 2.4. Determination of the Total Polyphenolic Content

The total content of the phenolic compounds in the extracts was determined in the reaction with the F–C reagent, as described earlier [25].

### 2.5. Statistical Analysis

Statistical analysis was performed using the Statistics 13.1 program (StatSoft, Cracow, Poland), and the significance was set at *p* < 0.05.

## 3. Results and Discussion

Polish honeys are very well-distributed in the Polish, as well as Eastern European, local markets, due to its high quality. As the first step, we evaluated the “false-positive” effect to eliminate the side effects of the spectrophotometric method, as proposed by Rhee et al. [26]. No false-positive effects in any honey studied in our work were observed.

### 3.1. Inhibition of AChE and BChE

The analysed honeys showed varied AChE and BChE inhibition (Figure 2 and Figure 3). The highest increase in terms of AChE inhibition (21.17%) was observed for thyme honey (number (no.) 13). A high inhibitive potential was also observed for honeys no. 8, 4, and 7 (respectively, 19.74%, 18.40%, and 18.31%). The lowest value (7.03%) was recorded for buckwheat honey (no. 5) (Figure 2).

In terms of BChE activity inhibition, the highest significant result was recorded for goldenrod honey—33.89% (no. 11) (Figure 3). The lowest inhibitive potential was observed for lavender honey—14.20% (no. 3).

In the past, only a limited number of papers have discussed the evaluation of the anti-ChE activity of honeys. Honey produced from Acacia trees significantly (*p* < 0.05), in a concentration-dependent manner, decreased AChE activity in the brain (cerebrum and cerebellum) and in male Wistar albino rat serum after 1 week of consumption (20% *v*/*v* Acacia honey, 5 mL/kg body weight) [27]. Philip and Mohd Fadzelly [28] reported on the anti-AChE and antioxidant activities of some honeys (young and old Mangrove, secondary forest, young and old Upper mountain, Tropical, and Potiukan honeys) from Malaysian Borneo. The highest inhibitory activity towards AChE was observed in the case of Manuka honey (produced from tea trees). The authors linked the high anti-ChE inhibitory activity of honeys with high polyphenolic contents. Philip and Mohd Fadzelly [28] correlated a high anti-AChE activity of the studied honeys with their high antioxidant activities, which can, in turn, be correlated with the high TPC in studied honeys. Zaidi et al. [29] detected significant anti-AChE activities of all 31 Algerian honeys, together with high TPC contents ranging from 14.50 to 99.62 mg GAE/100 g of honey. These authors pointed out a statistically confirmed, very high correlation between anti-AChE activities and TPC of studied honeys [29]. Additionally, Baranowska et al. [22] confirmed the positive correlation between anticholinesterase activity and TPC in a study on 47 samples of Polish honeys. They demonstrated that enzyme inhibition (AChE and BChE) depended on the flower from which the given honey was obtained, with the highest level of AChE inhibition (39.51%) observed for buckwheat honey, while the lowest (1.09%) was for multifloral honey. In the case of BChE, the highest significant inhibition (39.76%) was recorded for multifloral honey, and the lowest (3.06%) for honeydew honey [22]. Sahin [30] analysed samples of sunflower honey obtained from four regions of Adana in Turney and from different wholesalers throughout the city. All the tested samples of sunflower honey showed anticholinesterase activity. They inhibited acetylcholinesterase and butyrylcholinesterase by, respectively, 28.33–34.86% and 27.57–39.75% at 10 mg/mL. In turn, sunflower honey purchased from the wholesalers returned the results of, respectively, 14.77–21.36 and 14.18–23.64% when tested at the same concentration.

Our research confirmed the significant correlation between anticholinesterase activity and TPC in the honey samples under investigation (Figure 4). The statistical analysis revealed a positive correlation between TPC and anti-AChE activities of the tested honeys (*r* = 0.33 for TPC/AChE; *r* = 0.48 for TPC/BChE, *p* < 0.05).

The confirmation of the anti-ChE activity measured in in vitro and in vivo studies is a must. Inhibitors of the cholinesterases present in honeys exhibit a positive effect in vivo, as essentially shown in the experiment involving young and aged Swiss albino mice. The pre-treatment of the animals with mixed Swarnabhasma and Swarnaprashana honeys (15 days) significantly improved their learning and memory and significantly decreased the AChE activity in the whole brain [31].

### 3.2. Total Phenolic Content (TPC)

The TPC ranged from 326.23 mg GAE/100 g (Heather honey, no. 7) to 46.49 mg GAE/100 g (Aciacia honey, no. 1) (Table 2). The diversity in the content of the TPC in honeys, as well as in the world, is large, due to botanical origins but, also, the geographical area, storage conditions, seasonal, environmental factors, etc. The TPC of various honeys ranged between 5 and 20 µg/g honey [32], 7 and 20 µg/g honey [33], and 0.2 and 24 µg/g honey [34]. In five different types of Yemeni honey, and in one American, Swiss, and Iranian honey, the TPC ranged from 56.32 to 246.21 mg catechin equivalent/100 g honey [35]. In two Malaysian Apis mellifera honeys (Gelam and Coconut), the TPC ranged from 21.4 ± 1.29 to 15.6 ± 1.05 µg/g honey, respectively [36]. In another work, the TPC in six honeys was from 15.21 ± 0.51 mg GAE/kg (Borneo tropical honey) to 48.63 ± 1.21 mg GAE/kg (Manuka honey) [37]. Habib et al. [38] evaluated the TPC to be 30.81–132.60 mg GAE/100 g honey. In the work of Belkhodja et al. [39], the TPC in four honeys from Algeria was 0.32, 0.4, 0.76, and 0.39 mg GAE/100 g of honey [39]. In another study, the highest TPC among 40 honeys (unifloral and multifloral) from various countries in the world was in Manuka honey (New Zealand) (235.50 ± 16.01 μg/g), and the lowest in honeys from China (40.00 ± 6.01 μg/g) [40]. Single and multifloral Irish urban and rural honeys (131 Irish honey samples from 78 locations in Ireland, including 124 samples of multifloral honeys (55 and 69 urban and rural locations, respectively)), contained 2.59–81.10 mg of GAE/100 g of honey (*n* = 124, mean 23.84 ± 13.07) [41].

## 4. Conclusions

Given the role of cholinesterases and the need to inhibit their activity in the treatment of neurodegenerative diseases, the presented results may prove valuable, and the analysed honeys may find potential applications in the development of new medicinal products to facilitate the treatment of AD.

Based on the results of the present and earlier studies, our team continued this interesting research topic. Other tests were currently performed (“scavenging” of 2,2′-azino-bis(3-ethylbenzothiazoline-6-sulfonic acid (ABTS*^•^) and 2,2-diphenyl-1-picrylhydrazyl (DPPH^•^), cupric reducing antioxidant capacity, ferric antioxidant power, hydroxyl radical antioxidant capacity; oxygen radical absorbance capacity and effect on superoxide dismutase, glutathione peroxidase, and catalase and glutathione reductase activity) and will be published in the near future.

In our opinion, the results are relevant for those who would like to consider honeys as possible sources of cholinesterase inhibitors. We plan to focus on just a few honeys to study them more deeply (isolation of inhibitors, structure-activity relationships, etc.). Additionally, the testing of selected honeys using an animal model is mandatory.

## Figures and Tables

**Figure 1 nutrients-14-02969-f001:**
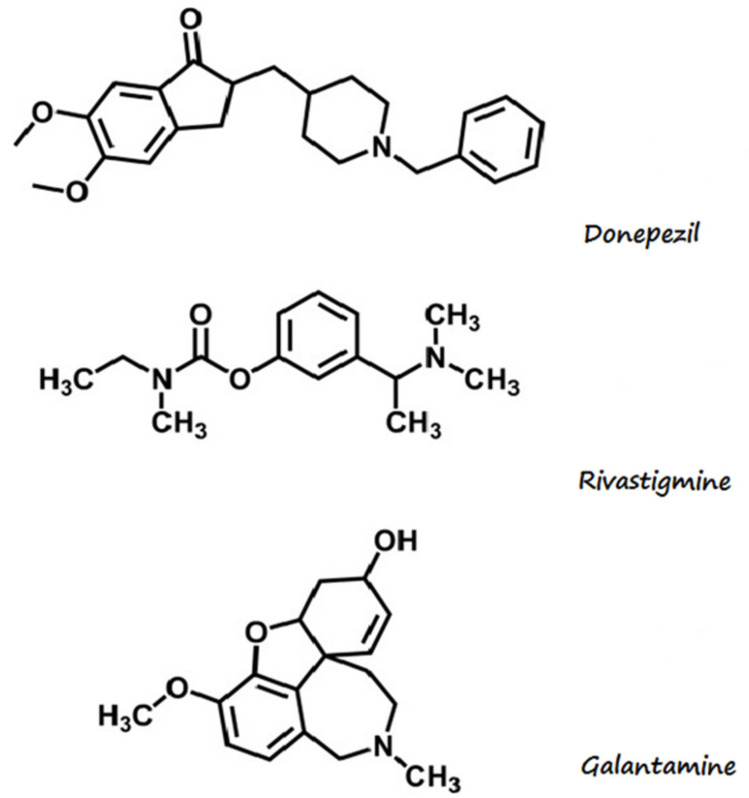
Structural formulas of cholinesterase inhibitors. Based on Nadri et al. [12]. O, oxygen; N, nitrogen; CH_3_, methyl group.

**Figure 2 nutrients-14-02969-f002:**
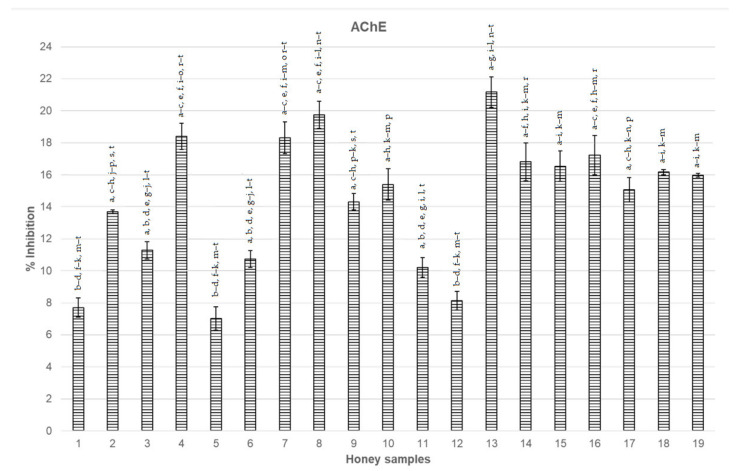
AChE inhibition (%) in the studied honeys (averaged results ± standard deviation). Legends: 1—Acacia, 2—Raspberry, 3—Lavender, 4—Bean, 5—Buckwheat, 6—Aloe, 7—Heather, 8—Linden, 9—Eucalyptus, 10—Sunflower, 11—Goldenrod, 12—Linden, 13—Thyme, 14—Rape, 15—Rosemary, 16—Hawthorn, 17—Pomegranate juice, 18—Acacia, and 19—Orange blossom. Various small letters a, b, etc. mean significant differences at *p* < 0.05. AChE, acetylcholinesterase.

**Figure 3 nutrients-14-02969-f003:**
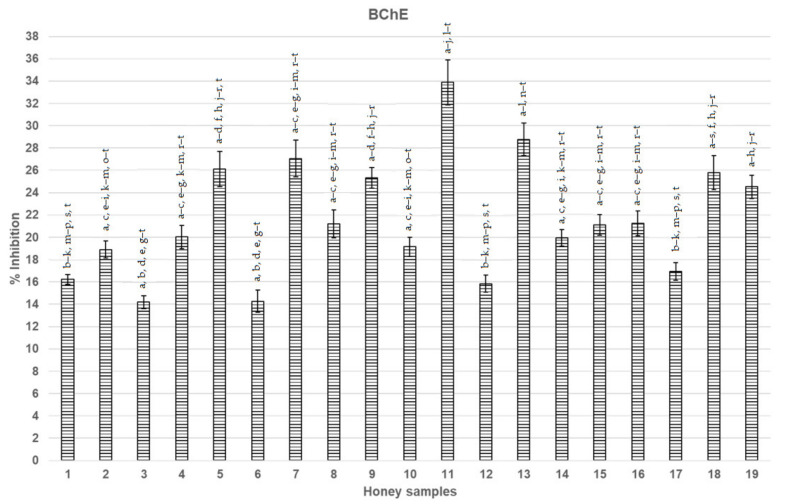
BChE inhibition (%) in the studied honeys (averaged results ± standard deviation). Legends: 1—Acacia, 2—Raspberry, 3—Lavender, 4—Bean, 5—Buckwheat, 6—Aloe, 7—Heather, 8—Linden, 9—Eucalyptus, 10—Sunflower, 11—Goldenrod, 12—Linden, 13—Thyme, 14—Rape, 15—Rosemary, 16—Hawthorn, 17—Pomegranate juice, 18—Acacia, and 19—Orange blossom. Various small letters a, b, etc. mean significant differences at *p* < 0.05. BChE, butyrylcholinesterase.

**Figure 4 nutrients-14-02969-f004:**
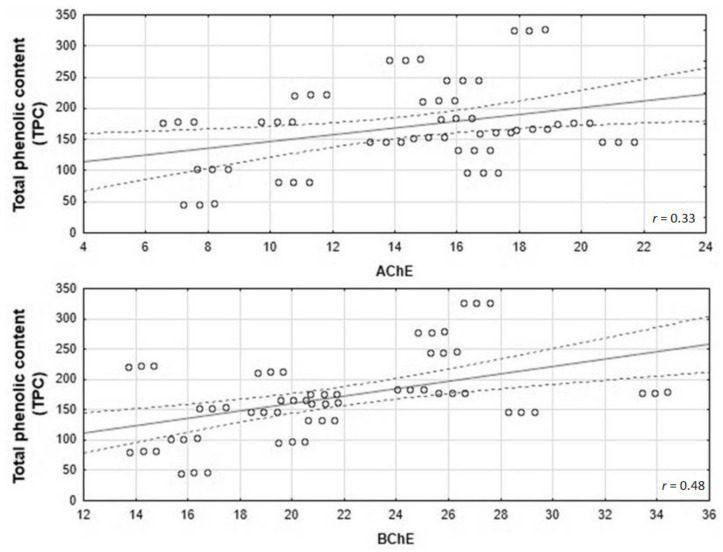
Correlation between anti-AChE activities and TPC of the studied honeys. R-Pearson’s correlation coefficient (*r*).

**Table 1 nutrients-14-02969-t001:** Samples of honey analysed.

Sample No.	Type of Honey	Country of Origin
1	Acacia	Poland
2	Raspberry	Poland
3	Lavender	Spain
4	Bean	Poland
5	Buckwheat	Poland
6	Aloe	Poland
7	Heather	Spain
8	Linden	Poland
9	Eucalyptus	Uruguay
10	Sunflower	Ukraine
11	Goldenrod	Poland
12	Linden	Poland
13	Thyme	Spain
14	Rape	Poland
15	Rosemary	Spain
16	Hawthorn	Poland
17	Pomegranate juice	Bosnia
18	Acacia	Poland
19	Orange blossom	France

No., number.

**Table 2 nutrients-14-02969-t002:** Total phenolic compound content (TPC) of the honeys analysed.

Honey Samples	TPC Content mg GAE/100 g
1	46.49 ± 3.30 ^b–t^
2	146.34 ± 10.86 ^a, c–l, n–t^
3	222.01 ± 9.45 ^a, b, d–t^
4	166.59 ± 8.64 ^a–d, f, g, i, j, l–t^
5	177.93 ± 23.66 ^d–e, g–t^
6	81.56 ± 11.78 ^d–e, g–t^
7	326.23 ± 18.05 ^a–f, h–t^
8	176.51 ± 33.69 ^a–d, f, g, i–t^
9	278.66 ± 16.32 ^a–h, j–t^
10	212.26 ± 10.82 ^a–i, k–t^
11	178.97 ± 18.32 ^a–d, f–j, l–t^
12	102.44 ± 20.32 ^a–k, m–t^
13	146.58 ± 7.49 ^a, c–l, n–t^
14	96.99 ± 10.15 ^a–m, o–t^
15	133.48 ± 9.69 ^a–n, p–t^
16	161.11 ± 5.24 ^a–o, r–t^
17	153.34 ± 14.00 ^a–p, s, t^
18	245.26 ± 3.09 ^a–r, t^
19	184.08 ± 8.64 ^a–s^

Various small letters a, b, etc. mean significant differences at *p* < 0.05. TPC, total phenolic compounds, GAE, gallic acid equivalents.

## Data Availability

The data that support the findings of this study are available from the corresponding author (E.B.-W.) upon reasonable request.

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
