# Peer review of "Honeys as Possible Sources of Cholinesterase Inhibitors"

_nutrients, 2022, doi:10.3390/nu14142969_

Round 1

Reviewer 1 Report

The manuscript has scientific sound.

Author Response

Let me thank you for your valuable comments concerning my paper.

Reviewer 2 Report

Research Article

Honeys as Possible Sources of Cholinesterase Inhibitors

This study aimed to evaluate the anti-AChE and anti-BChE activity and the total polyphenolic content (TPC) of 19 honeys.

-          This study well designed and written, but still some points need to be clarified.

-          Authors only depend on determinations of Cholinesterase Inhibitors in addition to measuring the total phenolic contents only.

-          We think also the determination of antioxidant activities is important and is being associated with the currently tested activities.

-          Also, authors did not determine what are the major compounds found among these different types of honey which could be responsible about the studied activity.

-          Or they can cite it from previous studies (antioxidant activities and the major compounds).

Author Response

This study aimed to evaluate the anti-AChE and anti-BChE activity and the total polyphenolic content (TPC) of 19 honeys.

-          This study well designed and written, but still some points need to be clarified.

AU: Thank you very much for your opinion.

-          Authors only depend on determinations of Cholinesterase Inhibitors in addition to measuring the total phenolic contents only.

AU: The health effects of honeys such as their antioxidant, anti-inflammatory properties or their use as an agent to prevent dementia are attributed mainly to the polyphenols present in honeys. Therefore, in this work, we wanted to focus on the evaluation of anti-AChE and anti-BChE activities to investigate their correlation with total polyphenol content (TPC), as we did similarly in our previous work (Baranowska-Wójcik, E., Szwajgier, D., & Winiarska-Mieczan, A. (2020). Honey as the Potential Natural Source of Cholinesterase Inhibitors in Alzheimer's Disease. Plant Foods for Human Nutrition. doi:10.1007/s11130-019-00791-1) to show whether they can indeed be considered as a possible source of cholinesterase inhibitors.

-          We think also the determination of antioxidant activities is important and is being associated with the currently tested activities.

AU: We agree with thisopinion. We performed a number of additional tests in (among others, we also performed antioxidant/reducing assays: ABTS and DPPH radicals’ “scavenging”,  cupric reducing antioxidant capacity, ferric antioxidant power, hydroxyl radical antioxidant capacity; oxygen radical absorbance capacity and effect on superoxide dismutase, glutathione peroxidase, catalase and glutathione reductase activity). These results are currently being prepared and will be published in next 2 papers. We added this sentence in the Conclusion, Line: 182-187

-          Also, authors did not determine what are the major compounds found among these different types of honey which could be responsible about the studied activity.

AU: As we have written above, detailed analyses of the tested honeys will be presented soon in our next work.

-          Or they can cite it from previous studies (antioxidant activities and the major compounds).

AU: The honeys are brought from different locations in Europe and are being tested by us for the first time, certainly detailed studies will be published by us soon.

Let me thank you for your valuable comments concerning my paper.

This manuscript is a resubmission of an earlier submission. The following is a list of the peer review reports and author responses from that submission.

Round 1

Reviewer 1 Report

From a scientific point of view, it is considered an excellent scientific research and adds scientific value to scientific research in this field, but it needs to be modified:
 1- There is a clear explanation for Figure No. (2) as it is not clear and requires an explanation of the results.

2-  Also  figs. ( 4   and  5  )  need  to  clarification  of  their  results.

3-  It  is   good  study  about  Cholinesterase Inhibitors.

4- There   is  no  clarification  about   Correlation between anti-AChE activities and TPC of studied honeys  in   fig. 4 .

5-  The  references  are  very  good  and  update.

Author Response

Manuscript ID: nutrients-1741366

Title: Honeys as possible sources of cholinesterase inhibitors

Nutrients

Reviewer: #1 Comments and Suggestions for Authors

From a scientific point of view, it is considered an excellent scientific research and adds scientific value to scientific research in this field, but it needs to be modified:

1- There is a clear explanation for Figure No. (2) as it is not clear and requires an explanation of the results.

AU: It was corrected as suggested.

2-  Also  figs. ( 4   and  5  )  need  to  clarification  of  their  results.

AU: It was corrected as suggested. But we aren't sure if that was the point. We think that by saying “figs. ( 4   and  5  )” you meant fig. 2 and 3?. If not, we are asking for more precise directions.

3-  It  is  good  study  about  Cholinesterase Inhibitors.

AU: Thank You??

4- There   is  no  clarification  about   Correlation between anti-AChE activities and TPC of studied honeys  in   fig. 4 .

AU: It was corrected as suggested. But we aren’t sure whether that was the point. If not, we are asking for more precise directions

5-  The  references  are  very  good  and  update

AU: Thank You??

Let me thank you for your valuable comments concerning my paper

Reviewer 2 Report

The manuscript presents an important topic for publication, but in the case of honey, despite the high number of samples, it is indicated that in the discussions the authors emphasize that honey undergoes profound changes in its composition according to numerous factors. So much so that the results of the different samples have already demonstrated this. But, they should have highlighted this aspect better.

In addition to this general detail, the work focused attention on the total phenolic compounds and, although the authors have presented that future work will be carried out, they would observe other important compounds in honey such as some amino acids, for example glutamic acid, which may be involved with these positive factors as a cholinesterase inhibitor or stimulator in other defense routes for this type of disease.

When it comes to the formatting of the manuscript, it is suggested that the authors note that volume units in “ml”, “mL”, “l”, “L”, isolated and in concentration units, must be standardized.

Example page 2:

“..Othman et al., (2015)..” Please correct to: “Othman et al. (2015)..”

Note errors of the same type throughout the text.

Page 3:

“..Folin-Ciocalteau (F-C)..”

Page 4:

“...with FC reagent..”

Please keep the abbreviation standards throughout the text.

Author Response

Manuscript ID: nutrients-1741366

Title: Honeys as possible sources of cholinesterase inhibitors

Nutrients

Reviewer #2 Comments and Suggestions for Authors

The manuscript presents an important topic for publication, but in the case of honey, despite the high number of samples, it is indicated that in the discussions the authors emphasize that honey undergoes profound changes in its composition according to numerous factors. So much so that the results of the different samples have already demonstrated this. But, they should have highlighted this aspect better.

In addition to this general detail, the work focused attention on the total phenolic compounds and, although the authors have presented that future work will be carried out, they would observe other important compounds in honey such as some amino acids, for example glutamic acid, which may be involved with these positive factors as a cholinesterase inhibitor or stimulator in other defense routes for this type of disease.

When it comes to the formatting of the manuscript, it is suggested that the authors note that volume units in “ml”, “mL”, “l”, “L”, isolated and in concentration units, must be standardized.

Example page 2:

“..Othman et al., (2015)..” Please correct to: “Othman et al. (2015)..”

Note errors of the same type throughout the text.

Page 3:

“..Folin-Ciocalteau (F-C)..”

Page 4:

“...with FC reagent..”

Please keep the abbreviation standards throughout the text.

AU: It was corrected as suggested.

Let me thank you for your valuable comments concerning my paper.